# Investigation of Nicotianamine and 2′ Deoxymugineic Acid as Enhancers of Iron Bioavailability in Caco-2 Cells

**DOI:** 10.3390/nu11071502

**Published:** 2019-06-30

**Authors:** Jesse T. Beasley, Jonathan J. Hart, Elad Tako, Raymond P. Glahn, Alexander A. T. Johnson

**Affiliations:** 1School of BioSciences, The University of Melbourne, Victoria 3010, Australia; 2Robert W. Holley Center for Agriculture and Health, USDA-ARS, Ithaca, NY 14853, USA

**Keywords:** biofortification, iron deficiency anemia, iron absorption, ferritin, ascorbic acid, epicatechin

## Abstract

Nicotianamine (NA) is a low-molecular weight metal chelator in plants with high affinity for ferrous iron (Fe^2+^) and other divalent metal cations. In graminaceous plant species, NA serves as the biosynthetic precursor to 2′ deoxymugineic acid (DMA), a root-secreted mugineic acid family phytosiderophore that chelates ferric iron (Fe^3+^) in the rhizosphere for subsequent uptake by the plant. Previous studies have flagged NA and/or DMA as enhancers of Fe bioavailability in cereal grain although the extent of this promotion has not been quantified. In this study, we utilized the Caco-2 cell system to compare NA and DMA to two known enhancers of Fe bioavailability—epicatechin (Epi) and ascorbic acid (AsA)—and found that both NA and DMA are stronger enhancers of Fe bioavailability than Epi, and NA is a stronger enhancer of Fe bioavailability than AsA. Furthermore, NA reversed Fe uptake inhibition by Myricetin (Myr) more than Epi, highlighting NA as an important target for biofortification strategies aimed at improving Fe bioavailability in staple plant foods.

## 1. Introduction

Iron (Fe) possesses unique redox properties that are critical to fundamental biological processes such as cellular respiration and photosynthesis [1]. Although abundant in soil, Fe is largely unavailable for plant uptake under aerobic or calcisol (high pH) conditions (representing ~30% of arable land), due to the formation of insoluble ferric (Fe^3+^) ion precipitates [2]. As well as negatively impacting on plant growth, inadequate plant Fe uptake translates to human Fe deficiency, as plants provide a major gateway for Fe into human food systems [3]. Plants have evolved sophisticated mechanisms to absorb Fe from the rhizosphere through reduction and/or chelation of Fe^3+^ [2]. Non-graminaceous plants such as common bean (*Phaseolus vulgaris* L.) reduce soil Fe^3+^ ions to the more soluble ferrous (Fe^2+^) form for uptake into plant roots [4]. By contrast, graminaceous plants such as bread wheat (*Triticum aestivum* L.) secrete mugineic acid phytosiderophores, the most common of which is 2′deoxymugineic acid (DMA), into soil to chelate Fe^3+^ for plant uptake [5]. Some plant species such as rice (*Oryza sativa* L.) utilize aspects of both strategies to maximize Fe uptake under a variety of soil and pH conditions [2].

Within the plant cell, Fe is complexed to chelating agents or is sequestered into plant vacuoles to avoid cellular damage caused by Fe^2+^ oxidation and reactive oxygen species (ROS) formation [3]. Low-molecular weight compounds like citrate, malate, nicotianamine (NA) and the oligopeptide transporter family protein (OPT3) are major chelators of phloem/xylem Fe within all higher plants while DMA is an additional chelator in graminaceous plants. Citrate, NA, DMA and OPT3 all function in the transport of Fe from source tissues (i.e., root, leaf) to sink tissues (i.e., leaf, seed) for Fe storage and/or utilization [4]. Within the leaf, most Fe is bound in a phytoferritin complex within the chloroplast [6]. Leaf Fe is liberated from the phytoferritin complex during senescence and chelated by citrate, NA and/or DMA for transport to the developing seed [4]. Once in the seed of non-graminaceous plants, the proportion of Fe stored in embryonic, seed coat, and provascular tissues is heavily influenced by species, genotype and environment [7,8]. The Fe within embryonic tissue is primarily bound to phytoferritin and represents between 18% to 42% of total seed iron in soybeans (*Glycine max* L.) and peas (*Pisum sativum* L.), respectively [9]. The Fe within the seed coat of common bean ranges between 4% and 26% of total seed iron and is bound primarily to polyphenolic compounds, such as flavonoids and tannins [8,10,11]. The majority of seed Fe therefore accumulates in cotyledonary tissues and is likely bound to inositol hexakisphosphate (also known as phytate) within cell vacuoles, or to small metal chelators like NA in the cytoplasm [7,12]. Certain leguminous plants like soybean and chickpea (*Cicer arietinum* L.) accumulate seed NA to very high concentrations (up to a 1:2 molar ratio with Fe), suggesting that a large proportion of seed Fe is cytoplasmic in these species [13,14]. Graminaceous plant seeds (i.e., grain) store the majority of Fe (~80% of total grain Fe) as phytate complexes in vacuolar regions of the outer aleurone layer [3,15,16]. The remaining Fe within the sub-aleurone and endosperm regions (~20% of total grain Fe) is bound to phytate in intracellular phytin-globoids or chelated to NA and/or DMA (1:0.1 molar ratio with Fe) within the cytoplasm [17,18,19,20].

The absorption of dietary Fe in humans (bioavailability) depends on several factors apart from Fe concentration alone. The Fe within plant-based foods is mostly comprised of low-molecular weight (i.e., phytate, NA) and high-molecular weight (i.e., ferritin) compounds and is collectively referred to as non-heme Fe [6]. Non-heme Fe bioavailability is generally low (5–12%) and influenced by the concentration of inhibitors (phytate, polyphenols, calcium, etc.) and enhancers, like ascorbic acid (AsA), in the diet [21,22]. Phytate is the major inhibitor of Fe bioavailability in whole-grain foods, although certain polyphenolic compounds such as myricetin (Myr) and quercetin exhibit a greater inhibitory effect in bean-based diets [10,21,22]. Both phytate and Myr form high affinity complexes with Fe^3+^ that are poorly absorbed across the human intestinal surface [23,24,25]. Other polyphenolic flavanoids present in wheat embryonic and bean seed coat tissues are widely presumed to inhibit Fe bioavailability through pro-oxidation of Fe^2+^ and/or chelation of Fe^3+^ [21,26,27]. Enhancers of Fe bioavailability such as AsA (the strongest enhancer identified to date) are typically antioxidants that reduce Fe^3+^ and prevent polyphenols binding to newly formed Fe^2+^ ions that are highly bioavailable [22]. Some polyphenols such as epicatechin (Epi) are also thought to reduce Fe^3+^ to Fe^2+^ and can therefore act as potent Fe bioavailability enhancers [21]. Another mechanism of promoting Fe bioavailability is thought to be through direct chelation of Fe^2+^ for uptake in the human small intestine such as that proposed for glycosaminoglycans and proteoglycans [22,28,29]. Nicotianamine has been suggested to enhance Fe bioavailability in Fe biofortified polished rice grains and Fe biofortified white wheat flour, although the extent of this promotion is unclear [17,18,30,31,32]. Whether DMA, also enhances or inhibits Fe bioavailability is unknown and increased knowledge regarding NA and DMA promotion of Fe bioavailability is needed. Identification of enhancers and inhibitors of Fe bioavailability has traditionally relied on manipulation of dietary components in lengthy human or animal feeding trials [33]. By contrast, the Caco-2 cell bioassay allows rapid investigation of diverse dietary components and accurate estimation of Fe uptake by human intestinal epithelial cells [21,34,35,36]. 

Due in large part to high consumption of cereal-based diets that are low in bioavailable Fe, human Fe deficiency is the most common nutritional disorder worldwide and is particularly widespread in less-developed countries [37]. Severe Fe deficiency causes iron-deficiency anemia (IDA), a condition that impairs cognitive development and increases maternal and child mortality, affecting over 40% of pregnant women and preschool-age children worldwide [38,39,40]. Biofortification efforts aimed at increasing micronutrient intake from staple food consumption represent a key component of alleviating global human IDA, yet there is heavy bias towards increased micronutrient concentration with less regard for bioavailability [41,42]. Recent biofortification studies in rice and wheat have increased NA and/or DMA biosynthesis to enhance both Fe concentration and bioavailability [17,20,28,29,43,44]. Increasing NA/DMA biosynthesis in wheat results in higher Fe accumulation in grain endosperm and increased Fe bioavailability in white flour that is highly correlated with NA and DMA concentration [17]. Understanding the extent to which NA and/or DMA enhance Fe bioavailability is therefore critical to determining the effectiveness of these Fe biofortification programs. Here we utilize modifications of the Caco-2 cell bioassay to characterize NA and DMA as enhancers of *in vitro* Fe bioavailability through comparison to known enhancers (Epi and AsA) and in a competitive assay with the inhibitor Myr.

## 2. Materials and Methods

### 2.1. Chemicals

Epicatechin, Myr, dimethyl sulfoxide (DMSO), glucose, hydrocortisone, insulin, selenium, triiodothyronine, and epidermal growth factor were purchased from Sigma-Aldrich (St. Louis, MO, USA). Nicotianamine and DMA were purchased from Toronto Research Chemicals Inc. (Toronto, Canada). Sodium bicarbonate and piperazine-N,N′-bis[2-ethanesulfonic acid] (PIPES) were purchased from Fisher Scientific (Waltham, MA, USA). Iron standard (1000 μg/mL in 2% HCl) was from High-Purity Standards (Charleston, SC, USA). Modified Eagle’s medium (MEM), Dulbecco’s modified Eagle’s medium (DMEM), and 1% antibiotic–antimycotic solution were purchased from Gibco (Grand Island, NY, USA).

### 2.2. Preparation of Metabolite and Fe solutions

Epicatechin and Myr were dissolved in DMSO (100%) to a concentration of 1.6 mM and NA and DMA were dissolved in DMSO (50%) and 18 MΩ H2O (50%) to a concentration of 0.8 mM. All solutions were diluted with pH 2 saline solution (140 mM NaCl, 5 mM KCl, adjusted to pH 2 with HCl) to achieve 400 μM stock solutions and subsequently diluted with pH 2 saline solution to appropriate concentrations for use in Caco-2 assays. To minimize toxicity to Caco-2 cells, the maximum DMSO concentration in 30 μM Epi/Myr treatments was 1.9% (2.5% in 40 μM polyphenol treatments). Fe stock solutions were prepared from 1000 mg/mL Fe standard in pH 2 saline solution. A 50 μL aliquot of Fe^2+^ stock solution of appropriate concentration was added to 150 μL of prepared metabolite solutions to achieve the desired Fe/metabolite concentration.

### 2.3. Caco-2 Assays

The Caco-2 cell assays were performed as previously described [21]. Briefly, cells were cultured in 24-well plates (Corning Costar 24 Well Clear TC-Treated Multiple Well Plates) coated with collagen and maintained in supplemented DMEM [3.7 g/L sodium bicarbonate, 25 mM HEPES (pH 7.2), 10% fetal bovine serum] for 12 days postseeding. Twenty-four hours prior to experiments, DMEM was replaced with iron-free supplemented MEM as previously described [36]. The Fe/metabolite solution (200 μL) was incubated (~22 °C, 15 min) and combined with 1 mL of MEM before an aliquot (500 μL) was directly applied to Caco-2 cell monolayers. After overnight incubation (37 °C), cells were washed twice with a buffered saline solution (130 mM NaCl, 5 mM KCl, 5 mM PIPES (pH 6.7)) and lysed by the addition of 0.5 mL 18 MΩ H2O. In an aliquot of the lysed Caco-2 cell solution, ferritin content was determined using an immunoradiometric assay (FER-IRON II Ferritin Assay, Ramco Laboratories, Houston, TX) and total protein content was determined using a colorimetric assay (Bio-Rad DC Protein Assay, Bio-Rad, Hercules, CA) as previously described [36]. As Caco-2 cells synthesize ferritin in response to intracellular Fe, we used the proportion of ferritin/total cell protein (expressed as ng ferritin/mg protein) as an index of cellular Fe uptake and refer to this as ‘Fe uptake’ throughout the manuscript.

### 2.4. Graphical Representation and Statistical Analysis

Each figure includes three control treatments as indicated by the white, grey and black bars on the right side of each panel. The first control treatment, “cell baseline”, represents ferritin formation in Caco-2 cells in the absence of any metabolite or Fe. The second control treatment, either “+ 4 μM Fe” or “+ 40 μM Fe”, represents ferritin formation in the presence of 4 μM Fe (typical Fe concentration for Caco-2 cell assays) or 40 μM Fe alone, respectively. A dotted line between both y-axes is provided to easily compare treatments to this control. The third control treatment, either “+ 80 μM AsA” or “+ 800 μM AsA”, represents ferritin formation in the presence of 4 μM Fe with 80 μM AsA or 40 μM Fe with 800 μM AsA (i.e., an Fe:AsA ratio of 1:20), respectively. In Figure 1, Figure 2 and Figure 3, triangular data points represent a fourth control treatment of ferritin formation in Caco-2 cells in the presence of NA, DMA, Epi or AsA solutions without Fe, and displayed values equivalent to the Cell Baseline. Data represent ng ferritin/mg protein in Caco-2 assays and were generated using SigmaPlot software (v.13.0, Systat Software, San Jose, CA, USA). Statistical differences between means were analyzed by unpaired Student’s t-test, and differences among means were assessed using one-way analysis of variance (ANOVA) with Tukey or Hsu’s MCB post-hoc tests, using Minitab software (v 18.0, Minitab, State College, PA, USA). 

## 3. Results

All low concentrations (≤2 μM) of NA, DMA and Epi enhanced Fe uptake into Caco-2 cells, with NA and DMA promoting ferritin formation more than Epi (Figure 1a). Higher concentrations (between 2–20 μM) of NA and Epi enhanced Fe uptake, and higher concentrations of DMA inhibited Fe uptake (Figure 1b). Between concentrations of 2 and 8 μM, NA enhanced Fe uptake more significantly than Epi with peak ferritin formation at 8 μM (i.e., a 1:2 molar ratio with Fe). At Fe molar ratios of 1:3, 1:4 and 1:5, NA and Epi promoted ferritin formation at the same level (Figure 1b and Appendix A). At a higher concentration of Fe (40 μM), NA enhanced Fe uptake significantly more than Epi at all concentrations apart from 2 μM, and DMA enhanced Fe uptake significantly more than Epi at 16 and 20 μM (Figure 2). Together, these results demonstrate that NA > DMA > Epi in the promotion of Fe uptake into Caco-2 cells. As the stronger enhancer, NA was compared in subsequent assays alongside Epi and AsA. All low concentrations (<2 μM) of NA, Epi and AsA enhanced Fe uptake into Caco-2 cells at similar levels, with NA showing significantly higher ferritin formation at 2 μM (Figure 3a). Above 2 μM, NA enhanced Fe uptake significantly more than both Epi and AsA with peak ferritin formation at 8 μM, demonstrating that NA > AsA > Epi in the promotion of Fe uptake into Caco-2 cells (Figure 3b). Across all experiments with Fe:metabolite molar ratios ≤ 1:2, the fold increase in ferritin formation over the Fe control was significantly higher in Caco-2 cells exposed to NA compared to AsA, Epi or DMA (Figure 4). To further characterize NA as a strong enhancer of Fe bioavailability, ferritin formation was measured in response to the Fe uptake inhibitor Myr in combination with NA (NA:Myr) or Epi (Epi:Myr). At 4 μM, total metabolite concentration, all NA:Myr solutions enhanced Fe uptake significantly more than Epi:Myr, and a NA:Myr solution of ratio 30:70 increased ferritin formation more than the Fe control (Figure 5a). At 30 μM total metabolite concentration, NA:Myr solutions of ratio 70:30, 80:20 and 90:10 enhanced Fe uptake significantly more than Epi:Myr, and a NA:Myr solution of ratio 90:10 increased ferritin formation more than the Fe control (Figure 5b). Together these results demonstrate that NA is stronger than Epi in counteracting the inhibitory effect of Myr and enhancing Fe uptake into Caco-2 cells.

## 4. Discussion

At low (≤1:2) Fe:metabolite molar ratios, NA and DMA enhanced Caco-2 cell ferritin formation more than Epi. Interestingly, the level of promotion was more pronounced at 40 μM Fe concentration compared to 4 μM Fe, despite maintaining the same Fe:metabolite molar ratios (Figure 1a and Figure 2). As ferritin formation was similar in the ‘Fe control’ at both 4 μM and 40 μM, it is likely that maximum Fe solubility is exceeded somewhere between 4 μM and 40 μM Fe in the absence of any Fe bioavailability enhancer (Figure 1 and Figure 2). Although we provide strong evidence to support NA and DMA as enhancers of Fe bioavailability, the exact mechanism of how NA and DMA facilitate Fe uptake into Caco-2 cells remains unknown and is likely dependent on the proportion of readily bioavailable ferrous Fe^2+^ ions in solution [21]. Both NA and DMA form high affinity 1:1 complexes with Fe^3+^ ions, however only NA is thought to be capable of binding Fe^2+^ [45,46]. Here we demonstrate that DMA promotes ferritin formation at Fe:DMA molar ratios less than 1:1, suggesting that DMA is capable of binding Fe^2+^ and/or reducing Fe^3+^ ions to some degree and requires further investigation. The decreased ferritin formation after exposure to Fe:DMA solutions with molar ratio > 1:1 (Figure 1b) could be due to oxidation of Fe^2+^ ions in solution and excess formation of Fe^3+^–DMA complexes that have low bioavailability for Caco-2 cell Fe uptake [45]. Some polyphenol compounds capable of complexing Fe^3+^ (e.g., delphinidin and delphinidin 3-glucoside) also demonstrate this biphasic pattern of promoting Fe uptake at molar ratios < 1:1 and inhibiting Fe uptake at molar ratios > 1:1 [21]. 

Nicotianamine promoted Caco-2 cell Fe uptake at all molar ratios observed, likely due to the formation of stable Fe^2+^–NA complexes and facilitation of Fe^2+^ uptake. As Fe is provided in a reduced Fe^2+^ state when preparing the assay, Fe^2+^–NA complexes rapidly form after NA addition and maintain the reduced Fe^2+^ state during exposure to Caco-2 cells. Certain polyphenols are also suggested to promote Caco-2 cell Fe uptake via a similar mechanism of binding Fe^2+^ and slowing the oxidation of Fe^2+^ to Fe^3+^ [21]. The reduced ferritin formation at Fe:NA molar ratios over 1:2 may be due to excess binding of Fe^2+^ by NA, preventing the release of Fe^2+^ for Caco-2 cell uptake, and suggests that an Fe:NA molar ratio less than 1:2 is optimal for promoting Fe bioavailability (Figure 1b, Appendix A). Alternatively, NA could enhance Fe bioavailability via direct uptake of Fe^2+^-NA in a similar mechanism to that proposed for glycosaminoglycans and proteoglycans and exploring this mechanism will be the subject of future research [28,29].

Ascorbic acid promotes Fe uptake into Caco-2 cells through *de novo* reduction of Fe^3+^ ions and maintenance of the Fe^2+^ ions in solution [9,21,22]. Here we demonstrate that at Fe:metabolite molar ratios less than 1:2, NA enhances Caco-2 cell ferritin formation significantly more than AsA and is the strongest enhancer of *in vitro* Fe bioavailability identified to date (Figure 3b and Figure 4). Together, these results suggest that the ability to bind Fe^2+^ ions is critical to enhancing Fe uptake into human intestinal cells. At an Fe:AsA molar ratio of 1:20, complete *de novo* reduction of Fe^3+^ ions in solution (without Fe^2+^ binding) leads to an ~8-fold increase in Caco-2 cell ferritin levels compared to Fe alone (Appendix A). An Fe:NA molar ratio of 1:20 was not tested as this ratio does not occur naturally in plant foods, although it is unlikely to show high ferritin formation given the effect of ratios greater than 1:2 (Appendix A). Instead, molar ratios less than 1:2 capture the highest ratio of Fe:NA measured to date in conventional plants foods (Fe:NA ratio of 1:1.6 in biofortified soybean) and provide realistic targets for plant breeders to improve Fe bioavailability [13].

To rule out the possibility that NA, DMA, Epi or AsA were promoting Caco-2 cell ferritin formation independent of Fe, Caco-2 cells were exposed to these metabolites alone in concentrations equal to 1.6 μM or 16 μM (Figure 1, Figure 2 and Figure 3). There was no sign that the metabolite presence alone significantly increased Caco-2 ferritin formation relative to the cell baseline or disrupted the Caco-2 cell monolayer at harvest. Thus, the increased ferritin formation observed in Caco-2 cells exposed to Fe:metabolite solutions is due to the Fe uptake promoting properties of these metabolites. 

Although Myr is a potent antioxidant with Fe-reducing properties (normally characteristic of Fe uptake enhancers), it strongly inhibits Caco-2 cell Fe uptake as it forms highly stable Fe–Myr complexes with low Fe bioavailability [47,48]. Nicotianamine increased ferritin formation compared to Epi at all 4 µM solutions and at 30 µM solutions containing over 70% NA or Epi, demonstrating that NA is stronger than Epi at preventing Fe uptake inhibition by Myr (Figure 5). At 30 µM, ferritin formation was equivalent to cell baseline for solutions containing less than 70% NA or 90% Epi, demonstrating that Myr inhibits Fe uptake more effectively than NA/Epi promotes Fe uptake (Figure 5b). At 4 µM (the equivalent of a 1:1 molar ratio with Fe), 30% NA combined with 70% Myr promoted ferritin formation above the Fe control, suggesting that NA can outcompete Myr and form Fe^2+^–NA complexes with high bioavailability (Figure 5a). An additional assay at 10 µM demonstrated an intermediate response, with low ferritin formation at 20% Myr due to improper Caco-2 cell growth (Appendix A).

Whether NA would enhance Fe bioavailability in the presence of several enhancers and inhibitors (i.e., a bean-based food sample) is unclear. To date, increased NA has not overcome the inhibitory effect of polyphenols/phytate to enhance *in vitro* Fe bioavailability within biofortified whole wheat grain [17]. To further address this question, future studies should explore NA biofortification within plant foods that contain high endogenous NA levels and low Fe:polyphenol molar ratios (e.g., 1:2 in carioca beans) [21]. Nonetheless, plant species with inherently low polyphenol levels (i.e., wheat) serve as ideal candidates for enhanced Fe bioavailability through the overproduction of NA [26,49]. The additional role of NA in the biosynthesis of DMA (itself an enhancer of Fe bioavailability) further reinforces increased NA biosynthesis as an effective cereal biofortification strategy to improve global human Fe nutrition.

## 5. Conclusions

We utilized a modified Caco-2 cell bioassay to characterize two low-molecular-weight plant metal chelators – NA and DMA – as strong enhancers of Fe bioavailability and demonstrate that NA is also capable of reversing Fe uptake inhibition by Myr. In doing so we highlight NA and DMA as important targets for biofortification strategies aimed at improving Fe bioavailability in cereals. Although we suspect that NA and DMA promote Fe bioavailability in Caco-2 cells by maintaining Fe^2+^ ions in solution, uncovering the exact mechanism by which these metal chelators promote Fe absorption will be the subject of future research studies.

## Figures and Tables

**Figure 1 nutrients-11-01502-f001:**
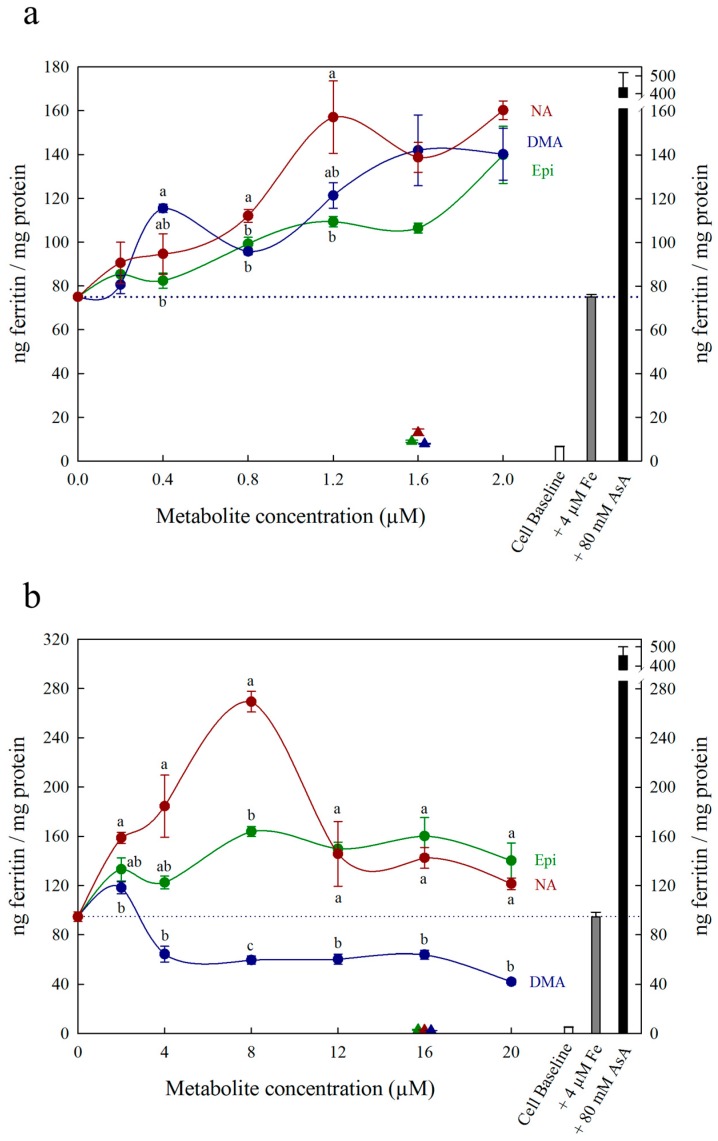
Ferritin formation in Caco-2 cells in response to nicotianamine (NA, red), 2′ deoxymugineic acid (DMA, blue) and epicatechin (Epi, green) at concentrations varying between **(a)** 0 and 2.0 μM and **(b)** 0 and 20 µM in solution with Fe (4 µM). Triangles indicate ferritin formation at metabolite solutions (1.6 µM or 16 µM) without Fe. Dotted line indicates ferritin response to 4 μM Fe alone and is extended to both y axes to facilitate comparison with other treatments. Error bars represent standard error of the mean of three replicates. Different letters indicate significantly different ferritin formation between metabolites of the same concentration as analyzed by one-way ANOVA with Tukey post-hoc test (*p* < 0.05).

**Figure 2 nutrients-11-01502-f002:**
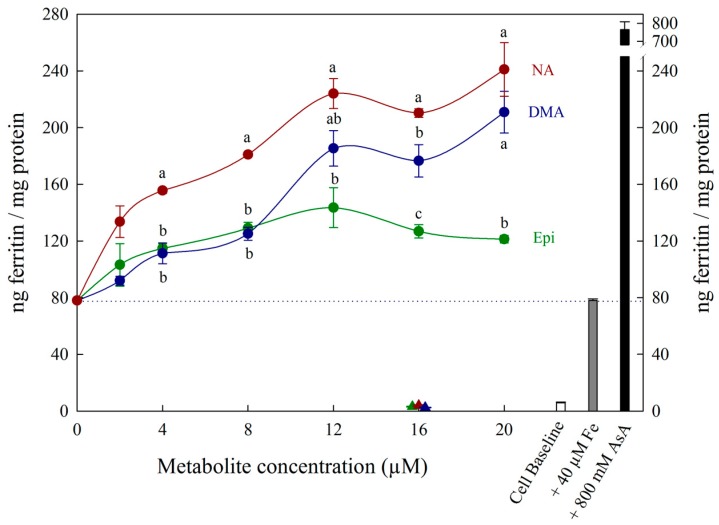
Ferritin formation in Caco-2 cells in response to varying concentrations (0–20 µM) of nicotianamine (NA, red), 2′ deoxymugineic acid (DMA, blue) and epicatechin (Epi, green) at concentrations varying between 0–20 µM in solution with Fe (40 µM). Triangles indicate ferritin formation in metabolite solutions (16 µM) without Fe. Dotted line indicates ferritin response to 40 μM Fe alone and is extended to both y axes to facilitate comparison with other treatments. Error bars represent standard error of the mean of three replicates. Different letters indicate significantly different ferritin formation between metabolites of the same concentration as analyzed by one-way ANOVA with Tukey post-hoc test (*p* < 0.05).

**Figure 3 nutrients-11-01502-f003:**
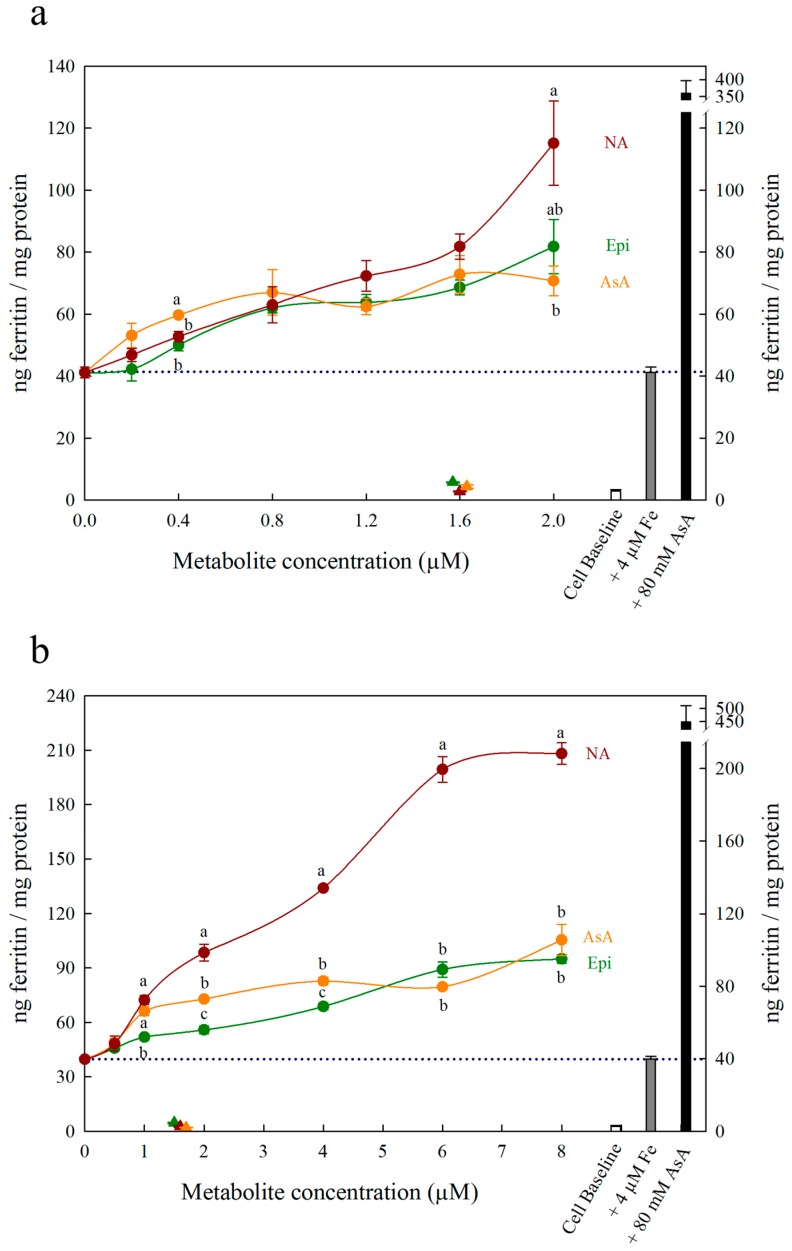
Ferritin formation in Caco-2 cells in response to nicotianamine (NA, red), ascorbic acid (AsA, orange) and epicatechin (Epi, green) at concentrations varying between (**a**) 0–2.0 μM and (**b**) 0–8 µM in solution with Fe (4 µM). Triangles indicate ferritin formation in metabolite solutions (1.6 µM) without Fe. Dotted line indicates ferritin response to 4 μM Fe alone and is extended to both y axes to facilitate comparison with other treatments. Error bars represent standard error of the mean of three replicates. Different letters indicate significantly different ferritin formation between metabolites of the same concentration as analyzed by one-way ANOVA with Tukey post-hoc test (*p* < 0.05).

**Figure 4 nutrients-11-01502-f004:**
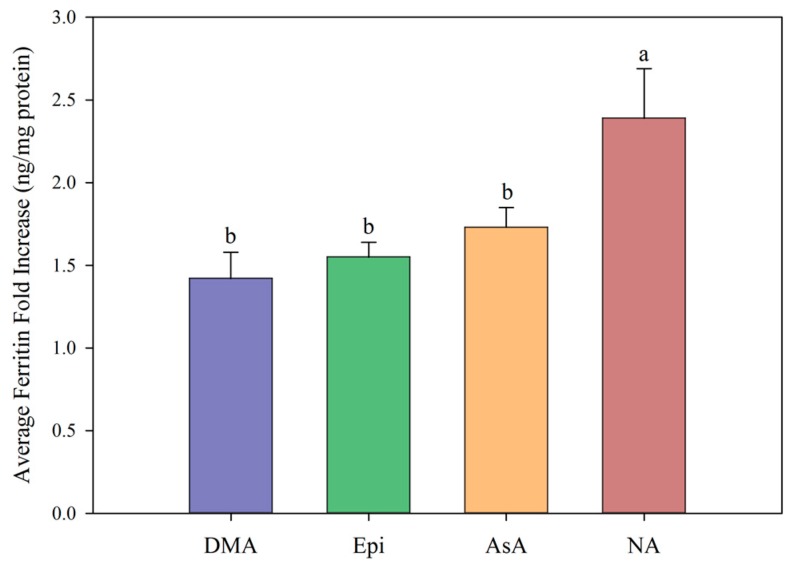
Average fold increase in Caco-2 cell ferritin formation in response to 2′ deoxymugineic acid (DMA, blue), epicatechin (Epi, green), ascorbic acid (AsA, orange), and nicotianamine (NA, red) at Fe:metabolite molar ratios ≤ 1:2 compared to ferritin formation in the presence of Fe alone. Error bars represent standard error of the mean of at least eight replicates. Different letters indicate significant differences between mean fold increase in ferritin formation between metabolites as analyzed by one-way ANOVA with Hsu’s MCB post-hoc test (*p* < 0.05).

**Figure 5 nutrients-11-01502-f005:**
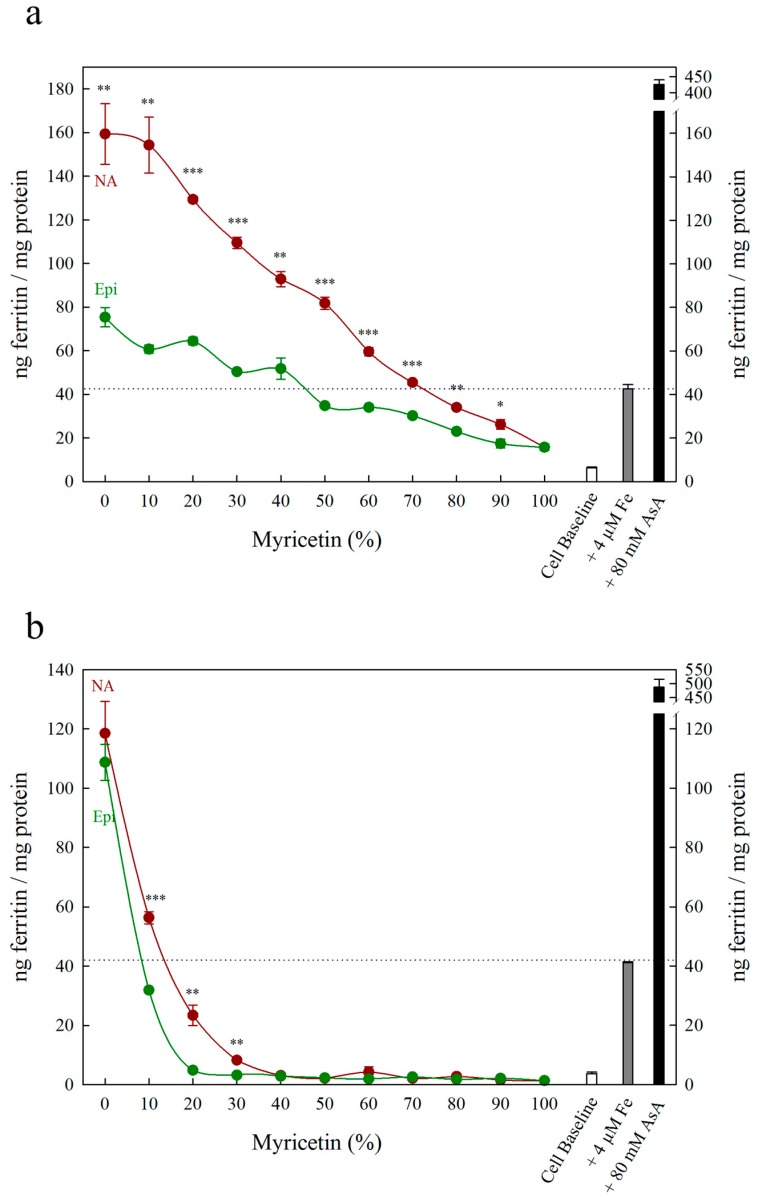
Ferritin formation in Caco-2 cells in response to various molar ratios of nicotianamine:myricetin (NA, red) and epicatechin:myricetin (Epi, green). Total metabolite concentration at each data point was (**a**) 4 µM and (**b**) 30 µM. Dotted line indicates ferritin response to 4 μM Fe alone and is extended to both y axes to facilitate comparison with other treatments. Error bars represent standard error of the mean of three replicates. Asterisks denote the significance between NA and Epi at each molar ratio for *p* < 0.05 (*), *p* ≤ 0.01 (**), *p* ≤ 0.001 (***) as determined by Student’s *t*-test.

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
