# Peer review of "Investigation of Nicotianamine and 2′ Deoxymugineic Acid as Enhancers of Iron Bioavailability in Caco-2 Cells"

_nutrients, 2019, doi:10.3390/nu11071502_

Round 1
Reviewer 1 Report
This novel analysis of Caco-2 cell response to Nicotianamine and 2' deoxymugineic acid found that both were stronger enhancers of iron bioavailability in this cell culture system compared to epicatechin and that NA was a stronger enhancer than ascorbic acid. NA further reversed inhibition of Fe uptake by myricetin more than epicatechin. These initial screening studies are important for plant breeders and research scientists working on biofortification strategies to improve the iron bioavailability of staple cereal crops.
Introduction is well-written.
Methods are thorough. Reference past methods.
Results are clearly explained in the text and graphically. All Figures are relevant. Supplementary Table and a Figure are referred to in Discussion.
Discussion is thorough and includes critical analysis of data.
Author Response
We thank the reviewer for their positive comments and have made no changes to the manuscript as a result of this review.
Reviewer 2 Report
This is a very good manuscript by Beasley et al. looking at the enhancing effect of nicotinamine and 2’deoxymugineic acid on iron uptake in Caco-2 cells. It has been written in a very clear and concise manner.
Minor comments:
1) Myr and Quercitin are known inhibitors of Fe uptake. What was the rationale for choosing Myr and not Quercitin?
2) Figure 4 only shows the error bars. Please upload this figure again.
3) DMA, Epi and NA only conditions were tested at 1.6uM and 16uM, could the authors please clarify why Myr:NA only and Myr:Epi only conditions were not tested.
4) The authors have proposed a possible alternative mechanism of uptake for Fe (Fe2+-NA) in the discussion (line 250), the authors may wish to cite some publications investigating the endocytic uptake of iron in Caco-2 cells.
Author Response
Comment 1
Myr and Quercitin are known inhibitors of Fe uptake. What was the rationale for choosing Myr and not Quercitin?
We did not make a change in relation to this comment. We have referenced a similar Caco-2 cell-based study in our manuscript that analyzed the inhibitory effect of polyphenols and found that “myricetin appears to be a more effective inhibitor than quercetin at a given concentration” [34].
Comment 2
Figure 4 only shows the error bars. Please upload this figure again.
We have addressed this comment by re-uploading Figure 4.
Comment 3
DMA, Epi and NA only conditions were tested at 1.6uM and 16uM, could the authors please clarify why Myr:NA only and Myr:Epi only conditions were not tested.
We did not make a change in relation to this comment. As exposure to either NA or Epi only did not increase cell ferritin formation above “Cell Baseline” (Figures 1, 2 and 3), we feel there is sufficient evidence that NA or Epi combined with Myr (Fe uptake inhibitor) would also not increase cell ferritin formation.
Comment 4
The authors have proposed a possible alternative mechanism of uptake for Fe (Fe2+-NA) in the discussion (line 250), the authors may wish to cite some publications investigating the endocytic uptake of iron in Caco-2 cells.
We have addressed this comment by citing two additional publications our Discussion section which now reads:
Alternatively, NA could enhance Fe bioavailability via direct uptake of Fe2+-NA in a similar mechanism to that proposed for glycosaminoglycans and proteoglycans and exploring this mechanism will be the subject of future research [45,46]. These references are now listed in our bibliography:
45. Laparra, J.M.; Tako, E.; Glahn, R.P.; Miller, D.D. Isolated Glycosaminoglycans from Cooked Haddock Enhance Nonheme Iron Uptake by Caco-2 Cells. J. Agric. Food Chem. 2008, 56, 10346–10351.
46. Huh, E.C.; Hotchkiss, A.; Brouillette, J.; Glahn, R.P. Carbohydrate Fractions from Cooked Fish Promote Iron Uptake by Caco-2 Cells. J. Nutr. 2004, 134, 1681–1689.